# Relationship between Nutritional Status and Clinical and Biochemical Parameters in Hospitalized Patients with Heart Failure with Reduced Ejection Fraction, with 1-year Follow-Up

**DOI:** 10.3390/nu12082330

**Published:** 2020-08-04

**Authors:** Marta Kałużna-Oleksy, Helena Krysztofiak, Jacek Migaj, Marta Wleklik, Magdalena Dudek, Izabella Uchmanowicz, Maciej Lesiak, Ewa Straburzyńska-Migaj

**Affiliations:** 11st Department of Cardiology, University of Medical Sciences in Poznan, 61-848 Poznan, Poland; marta.kaluzna@wp.pl (M.K.-O.); protozoan@o2.pl (J.M.); magdamroz8@gmail.com (M.D.); maciej.lesiak@skpp.edu.pl (M.L.); ewa.straburzynska-migaj@skpp.edu.pl (E.S.-M.); 2Poznan University of Medical Sciences Hospital of Lord’s Transfiguration, 61-848 Poznan, Poland; 3Faculty of Health Sciences, Wroclaw Medical University, 50-367 Wroclaw, Poland; marta.wleklik@gmail.com (M.W.); izabella.uchmanowicz@eckp.wroclaw.pl (I.U.)

**Keywords:** heart failure, reduced ejection fraction, nutritional status, malnutrition, mortality

## Abstract

Heart Failure (HF) is a cardiovascular disease with continually increasing morbidity and high mortality. The purpose of this study was to analyze nutritional status in patients diagnosed with HF with reduced ejection fraction (HFrEF) and evaluate the impact of malnutrition on their prognosis. The Polish version of MNA form (Mini Nutritional Assessment) was used to assess the patients’ nutritional status. The New York Heart Association (NYHA) class, exacerbation of HF, chosen echocardiographic and biochemical parameters, e.g., natriuretic peptides or serum albumin, were also analyzed. Among the 120 consecutive patients, 47 (39%) had a normal nutritional status, 62 (52%) were at risk of malnutrition and 11 (9%) were malnourished. The patients with malnutrition more frequently presented with HF exacerbation in comparison to those with normal nutritional status (82% vs. 30% respectively, *p* = 0.004). There were no significant differences between the investigated groups as to natriuretic peptides; however, both the malnourished patients and those at risk of malnutrition tend to show higher B-type natriuretic peptide (BNP) and NT-proBNP concentrations. During the average 344 days of follow-up 19 patients died and 25 were hospitalized due to decompensated HF. Malnutrition or being at risk of malnutrition seems to be associated with both worse outcomes and clinical status in HFrEF patients.

## 1. Introduction

Heart failure (HF) is a cardiovascular syndrome with a still-high mortality [1]. Poor prognosis applies especially to the patients with HF in conjunction with reduced ejection fraction (HFrEF). The most prevalent etiology of HF varies worldwide, depending on the region. Its classification remains unclear, because there is a long list of factors that can cause HF, such as ischemic heart disease, cardiomyopathies, hypertension [2], or cytostatic drugs administered in oncology [3].What is more, many patients will simultaneously develop several comorbid conditions and risk factors leading to HF. It is clear that elevated blood pressure, obesity, diabetes mellitus [4], and dyslipidemia [5] contribute to the development of HF.

HF is a growing burden for health care systems because its incidence is constantly increasing. The number of hospitalizations due to HF is rising each year, and it has tripled over the last 3 decades. HF is responsible for a large proportion of deaths as well as for diverse morbidity, which translates into the reduced quality of life in patients with HF [6]. Both in-hospital and 1-year outcomes of patients admitted with acute HF are still not satisfactory, due to particularly high mortality rates [7].

Chronic HF (CHF) is a complex syndrome resulting from a heart injury, which in turn causes injury of other organs. It is often accompanied by malnutrition, which is associatedwithan imbalance between the catabolic and the anabolic processes. This eventually results in cardiac cachexia, which worsens the prognosis of HF patients [8,9]. The mechanisms underlying malnutrition in HF are still under investigation, but several hypotheses have been developed. Some authors consider ventricular dysfunction, more likely of the right than of the left ventricle [10], a trigger of intestinal edema that is likely to induce activation of proinflammatory cytokines resulting in malabsorption and malnutrition [11]. Others explain malnutrition in HF patients as a result of intestinal microbiota imbalance or intestinal epithelium dysfunction, which can impair absorption of nutrients [12]. Disruption of intestinal barrier may contribute to developing malnutrition, but further research is needed to confirm this hypothesis [13]. As early as in 1997 it was observed that cachexia seemed to be an independent risk factor for HF patients’ mortality [8]. Inappropriate nutritional status also appears to be associated with the disease’s severity and progress [14]. As a matter of fact, malnutrition among HF patients is common [15,16], which prompted the European Society of Cardiology to indicate the importance of preventing malnutrition in HF patients in its guidelines for the management of HF [2]. Nevertheless, there is still no gold-standard for the diagnosis and evaluation of malnutrition in HF patients. Finding the HFrEF patients at risk of malnutrition is the key to timely employment of treatment and preventing its development.

## 2. Materials and Methods

### 2.1. Participants

This prospective single-center study of patients diagnosed with HFrEF enrolled 120 consecutive adult patients hospitalized at the 1st Department of Cardiology of Poznan University of Medical Sciences between 20 June 2017 and 10 December 2018. The analyzed group included both patients admitted to hospital due to stable and decompensated CHF. The patients were classified according to the international statistical classification of disease (ICD-10) for the diagnosis of HF (I50). The inclusion criteria included: (1) admission to the department of cardiology due to CHF (ICD-10 code for main diagnosis I50); (2) age ≥ 18 years; (3) HF history longer than three months; (4) left ventricular ejection fraction (LVEF) < 40%; (5) signing the informed consent form to enroll in the study. The research was conducted in compliance with the Declaration of Helsinki and accepted by the Ethics Committee of Poznan University of Medical Sciences (approval code 926/14).

### 2.2. Clinical Assessment

On admission, detailed medical history was collected. Special attention was paid to other conditions that might influence the patient’s nutritional status (e.g., neoplasms) and comorbidities modifying the cardiovascular risk, like diabetes mellitus (DM), chronic kidney disease (CKD), chronic obstructive pulmonary disease (COPD), hypertension, or previous myocardial infarction (MI). The patients were classified according to the New York Heart Association (NYHA) functional classification as indicated by the European Society of Cardiology (ESC) guidelines [2]. They also underwent a physical examination, including measurements of blood pressure (BP), heart rate (HR), height, and body mass. The patients were weighed without shoes and with light clothes on, with the use of a standardized and controlled weight with a digital scale to the nearest 0.1 kg. The following formula was used to calculated body mass index (BMI): BMI = weight (kg)/height (m)^2^ [17]. ECG was performed in all patients to assess their heart rhythm. Fasting blood samples were taken in the morning. The blood tests of particular interest were complete blood count, natriuretic peptides (particularly B-type natriuretic peptide (BNP)and N-terminal pro B-type natriuretic peptide (NT-proBNP), lipid panel, creatinine, fasting glucose, serum albumin, aspartate aminotransferase (AST), alanine aminotransferase (ALT), thyroid hormones and electrolytes (sodium, potassium), C-reactive protein (CRP). LVEF was assessed in echocardiography using the Simpson method (according to the guidelines [2]).

### 2.3. Nutritional Screening

The evaluation of the patients’ nutritional status was based on the Polish version of the MNA (Mini Nutritional Assessment) form (provided by Société des Produits Nestlé SA 1994, Revision 2009, Vevy, Switzerland, Trademark Owners, which holds the copyright of the instrument: http://www.mna-elderly.com/). The MNA questionnaire is a simple, noninvasive tool to assess and demonstrate malnutrition, validated over 25 years ago [18,19,20]. This questionnaire was developed for evaluation of the elderly [18,19,20,21,22]. In clinical practice, two versions of this questionnaire are used: a full version developed in 1994 [19] and a short form known as the MNA-SF version [23,24]. Because of clearly defined thresholds it is commonly used by clinicians in their daily practice all over the world [21]. It consists of 18 questions and 2 parts: Screening (6 questions) and Assessment (12 questions). For every answer the patients get points, which are summed up at the end. The first part includes questions about loss of appetite, mobility, weight loss during the last 3 months or neuropsychological problems and BMI evaluation. The second part involves questions related to a patient’s diet (number of meals, food, and fluid intake), number of drugs taken, self-assessment of nutritional status, and general condition. In addition, calf and mid-arm circumferences are measured. Malnutrition indicator score is a sum of the screening score (max. 14 points) and the assessment score (max. 16 points), and a patient may scorea maximum of 30 points. The patients are divided into 3 subgroups according to the final number of points: normal nutritional status (24–30 points), at risk of malnutrition (17–23.5 points), and malnourished (fewer than 17 points). A low MNA score was proved to be associated with a longer duration of hospitalization and an increased mortality [18]. This study analyzed the nutritional status, clinical condition, and biochemical parameters of the patients, and the frequency of all-cause readmissions, rehospitalizations due to cardiovascular diseases and cardiovascular and all-cause mortality during a one-year follow-up.

### 2.4. Statistical Analysis

Statistical analysis was performed using STATISTICA 13 Tibco Software Inc., Palo Alto, CA, USA. Probability distribution of continuous variables was tested with Lillefors and Shapiro-Wilk tests, and the variables were found to have non-normal distribution. Hence, Mann-Whitney U and Kruskal-Wallis ANOVA tests were used for the continuous variables. Chi square tests were used for the categorical variables. Univariate regression models, log rank tests, and Kaplan-Maier plots were used to assess the unadjusted survival. Multivariate analysis of survival was performed using Cox proportional hazard regression models with the adjustment for the parameters that significantly differed between the survivors and the patients who died, and the additionally nutritional status. The data is expressed as mean values with standard deviation for the continuous variables and percentages for the categorical variables. A *p* value of <0.05 was considered statistically significant for all the tests.

## 3. Results

The study sample consisted of 120 consecutive patients with HFrEF, 17% of whom were women. In total, 53 patients were hospitalized due to CHF exacerbation and 67 for CHF evaluation. The median age was 58 years (mean 55 ± 11 years). In total, 3% of the patients presented NYHA class I, 35% NYHA class II, 49% NYHA class III, and 13% NYHA class IV. The median LVEF was 25%, mean 26 ± 11%. At the time of inclusion in the study 98% of the patients were treated with beta-blockers, 70% with angiotensin-converting enzyme inhibitors (ACEI), 9% with angiotensin receptor blockers (ARB), 7% with sacubitril/valsartan, and 90% with mineralocorticoid receptor antagonists (MRA).

Among the patients enrolled to the study, 47 (39%) had normal nutritional status (group 1), 62 (52%) were at risk of malnutrition (group 2), and 11 (9%) were malnourished (group 3). None of the patients were treated with nutritional supplements. There were no significant differences in gender and age, as well as in etiology of HF (ischemic or non-ischemic) between the groups with different nutritional status. The difference in length of index hospitalization duration between the three groups was not significant (Table 1 and Table 2), although the data seems to show a certain pattern that most likely requires verification in a larger group of patients. The difference in length of index hospitalization duration is significant when the patients with normal nutritional status are compared with the combined groups of malnourished patients and those at risk of malnutrition (Table 3 and Table 4).

The Table 1 and Table 2 show the characteristic for patients with normal nutritional status, those at risk of malnutrition, and the malnourished patients at the time of enrollment in the study (on admission to hospital). The patients with different nutritional status also differed in many other ways. It seems remarkable that NYHA class on admission was higher in malnourished patients in comparison with the other groups (*p* = 0.008). However, the risk of malnutrition appeared independently of the NYHA class, and the patients at risk of malnutrition were classified as NYHA I to IV. Among the malnourished patients only one person presented with a lower NYHA class (I or II) and the majority presented with higher functional classes (NYHA III or IV). The patients with malnutrition were much more likely to present with exacerbation of CHF in comparison with the patients with normal nutritional status (82% vs. 30%, respectively, *p* = 0.004). BMI was significantly lower in malnourished patients than the patients with normal nutritional status, though it was still within the normal range (22.2 ± 6.6 kg/m^2^ vs. 28.8 ± 4.9 kg/m^2^, respectively; *p* < 0.001). The malnourished patients also demonstrated lower systolic BP on admission (*p* = 0.018), higher heart rate on discharge (*p* = 0.009), smaller distance in 6-min walk test (6MWT) (*p* = 0.031), lower maximum oxygen consumption during cardiopulmonary exercise test (VO_2_ L/min; *p* = 0.0159), and higher VE/VCO_2_ slope (*p* = 0.001). When the guideline-based treatment for HFrEF was analyzed, malnourished patients appeared to receive ACEI (angiotensin-converting enzyme inhibitors) or ARB (angiotensin receptor blockers) or ARNI (sacubitril/valsartan) less frequently than the patients with normal nutritional status. A similar trend was observed when we compared the combined group of patients at risk of malnutrition and malnourished to the patients with normal nutritional status (Table 4). There were no significant differences in use of beta-blockers and MRA (mineralocorticoid receptor antagonists).

Total cholesterol level was significantly lower in the malnourished patients in comparison with the patients with a normal nutritional status (3.61 ± 0.9 mmol/L vs. 4.67 ± 1.61 mmol/L; *p* = 0.021), as well as LDL-cholesterol (LDL-C) (1.9 ± 0.5 mmol/L vs. 2.65 ± 1.3 mmol/L; *p* = 0.014), they also had lower fasting glucose concentrations. There were no significant differences between the malnourished patients and those with normal nutritional status as to hsCRP and natriuretic peptides, neither BNP nor NT-proBNP; however, both the malnourished patients and those at risk of malnutrition tend to show higher concentrations of these particles, which requires verification in a larger group of patients. There were also no significant differences between the groups as to total protein and albumin concentrations. When comparing the patients with normal nutritional status and the combined groups of the malnourished patients and those at risk of malnutrition it appears that the latter show significantly higher NT-proBNP, lower maximum oxygen consumption during exercise, and higher prevalence of diabetes (Table 3 and Table 4).

During the average 344 days of follow-up (maximum 619 days, median 421 days), 19 (16%) patients died from all the causes and 25 (21%) were rehospitalized due to HF exacerbation. Mortality rate was 15% in patients with normal nutritional status, 15% in those at risk of malnutrition, and 37% in the malnourished patients. The non-survivors showed longer index hospitalization time (8 vs. 6.9 days; *p* = 0.042), lower systolic blood pressure on admission (99 mmHg vs. 110 mmHg; *p* = 0.019), lower eGFR (56 mL/min/1.73 m^2^ vs. 73 mL/min/1.73 m^2^; *p* = 0.007), lower albumin concentration (32.6 g/L vs. 47.4 g/L; *p* < 0.001), lower concentration of total cholesterol (3.4 mmol/L vs. 4 mmol/L; *p* = 0.007), higher NYHA class (3.2 vs. 2.6; *p* = 0.011), higher VE/VCO_2_ slope (41 vs. 34; *p* = 0.036), higher TSH concentration (5.16 uIU/mL vs. 2.51 uIU/mL; *p* = 0.013), and higher NT-proBNP concentration (5574 pg/mL vs. 2789 pg/mL; *p* = 0.016). The non-survivors were also more likely to present with exacerbation of HF (68% vs. 40%; *p* = 0.02) and more often suffered from chronic kidney disease (53% vs. 16%; *p* < 0.001) and thyroid disorders (53% vs. 18%; *p* = 0.001). They did not differ significantly from the survivors as to the nutritional status.In this study only albumin concentration and eGFR proved to have a prognostic value. The nutritional status influenced survival during the follow-up neither in single-variate (Figure 1) nor in multivariate analysis (Table 5).

## 4. Discussion

Malnutrition in patients suffering from HF remains an underestimated issue, even though the prevalence of abnormal nutritional status is common among these patients. The exact prevalence of malnutrition is difficult to evaluate because of lack of standardized methods of diagnosis. In hospitalized patients with CHF, the prevalence of nutritional risk differs from 34% to 90%, depending on the employed screening tools and the investigated population [16,25,26,27,28]. Our study shows malnutrition in 9%, risk of malnutrition in 52%, and normal nutritional status in 39% of hospitalized patients, which is similar to the Spanish study by Bonilla-Palomas et al. [25]. Overall, there were 61% of patients with abnormal nutritional status in our study. It is difficult to assess the nutritional status and initiate effective treatment, particularly in decompensated HF patients. To diagnose malnutrition, we typically use the MNA form in our Institution, described in detail in Methods. Nonetheless, we are aware it is not an ideal method in this group of patients (e.g., assessing BMI, calf, or mid-arm circumferences in patients with oedema is deemed contentious). In our study none of the patients were treated with nutritional supplements but we cooperated with a dietitian, who is responsible for helping the patients understand their health and nutritional challenges and is trying to resolve their problems with the dietary interventions. In the absence of comprehensive nutritional guidance for HF patients, it appears that small increases in energy, protein (red meat), and vegetable consumption are associated with improved nutritional status, which may prevent adverse events in this population. Such positive changes have been shown in a secondary data analysis of survey data from the National Health and Nutrition Examination Survey (NHANES) III using a cross-sectional design [29]. Attempts are being made to systematize the management of a patient with HF after being discharged from the hospital. The use of an algorithm-based, personalized discharge checklist (PCL) was associated with a significant higher referral to follow-up programs, better screening, and treatment of malnutrition and iron and vitamin D deficiencies in patients hospitalized for acute HF [30]. However, there were no significant changes in outcomes during the 6-month follow-up period [30]. All in all, there is no doubt nutritional status assessment is essential in HF patient management. Micronutrient deficiencies and borderline status are more common than generally acknowledged [31]. The most important acute deficiency being thiamine level [31]. The above is confirmed by Bilgen et al. who claims that inpatients dietary assessment and proper nutritional interventions may decrease readmission rate in patients with HF and improve their quality of life [32].

Some authors state that patients at high risk of malnutrition are also at high risk of cardiovascular death (e.g., because of myocardial infarction, decompensation of CHF or sudden cardiac death) [16]. That is why it is so important to find reliable screening tools and biomarkers of abnormal nutritional status to identify the patients at risk, and thus prevent malnutrition and cardiac cachexia.

There is research which also demonstrates that CONUT score (controlling nutritional status) improved the risk prediction of adverse events compared to traditional risk factors in coronary artery disease (CAD) patients after percutaneous coronary intervention (PCI) [33]. Analyzing a group of patients with CAD undergoing PCI Chen et al. revealed that inappropriate nutritional status is associated not just with higher risk of future acute myocardial infarction or developing congestive heart failure, but also cardiovascular death [33].

According to Sze et al. [15], who analyzed occurrence and prognostic value of malnutrition in HF patients, malnutrition appeared more frequently in patients with high NT-proBNP level (>4000 ng/L). This is not confirmed by our results, even though the malnourished patients and those at risk of malnutrition showed a trend towards higher NT-proBNP. In addition, Sze et al. showed that a worse nutritional status seems to be associated with worse outcomes independently of LVEF [15]. In that research, 3 different malnutrition scores (PNI (prognostic nutritional index), GRNI (geriatric nutritional risk index) and CONUT (controlling nutritional status)) were used and none of them was MNA form [15]. Our study, based on MNA form, did not show any influence of malnutrition on the risk of death. The study of Sze et al. [15] includes both the patients with HFrEF and HFpEF, and only 35% of the patients had LVEF <40%. They analyzed the prognosis for both groups combined, so the population was completely different than in our study. The patients were also much older than in our study, with a mean age of 75 years (>20 years more than our patients), and the youngest patient was 64 years old. Different tools for diagnosing malnutrition and an older population likely resulted in the demonstrated influence of malnutrition on the prognosis. Our study, based on a more homogeneous and younger population of patients, did not confirm this finding.

There are studies showing no significant differences in in-hospital mortality and the length of hospital stay between patients with GNRI <92 and ≥92, among HFrEF patients with acute decompensated HF [34]. The above results indicate the assessment of nutritional status with GNRI as useful for stratifying patients at high risk for longer length hospital stays in HFpEF but not in HFrEF [34]. In our study the assessment of nutritional status with the use of the MNA form demonstrated that nutritionally sound patients were hospitalized for shorter periods of time than those at the risk of malnutrition and the malnourished, analyzed together (8.4 days vs. 5.4 days; respectively).

There are also findings indicating that nutrition may play a pivotal role in metabolic protection in the HFpEF population [35], which was not covered in our study, devoted exclusively to HFrEF.

A Spanish study investigating impact of malnutrition on long-term mortality in patients diagnosed with chronic HF also underlined the fact that malnutrition is an independent predictor of mortality in HF patients [36]. The assessment of malnutrition in this research was based on the MNA form and the median follow up was 28 months. However, the analyzed population was much older than in our study (the mean age in that group was 74.6 ± 10.1 years) [36]. The study of Aggarwal et al. [26] showed malnutrition to be an independent predictor of mortality in patients with advanced HF. This study enrolled patients with severe HF qualified for left ventricular assist device (LVAD) implantation or heart transplantation (HTx). This was contrary to our study, in which just 15% of the analyzed patients were qualified for HTx: 7 (19%) patients in the group with normal nutritional status, 6 (12%) at risk of malnutrition, and 2 (33%) with malnutrition, and this likely influenced mortality in the malnourished group. In our study, 43% of combined malnourished patients and those at risk of malnutrition died during the 1-year follow-up. This is consistent with the previous studies where the mortality was found to be between 35.9% and 76% [25] or between 26.5% and 42% [26] in HF patients with poor nutritional status (combining those at risk of malnutrition and the malnourished). Both studies used the MNA questionnaire as the screening tool [25,26]. Another study using a different screening tool (Nutritional Risk Screening; NRS-2002) had shown even higher mortality during a three year follow-up (73.9% of CHF patients who were classified as severe malnutrition) [37]. It should be underlined that the follow-up in these studies was much longer: 2 [25] and 3 years [26,37] compared with the median follow-up in our study: 421 days. If the follow-up of our patients were longer, the results could have been different—probably more in line with the results of the other authors. Age is another important factor influencing mortality, and it was noticeably lower in our study (56.2 ± 11 years) than in the studies by Tevik et al. (median age 78 years, range 37–95 years) [37] and Bonilla-Palomas et al.’s research (mean age 73 ± 10 years) [25]. The mean age of the analyzed group in the study by Aggarwal et al. (59.3 ± 14.1 years) [26] is similar to ours, but they investigated a population of patients with advanced HF qualified for LVAD therapy or HTx. By definition, such patients demonstrate poor prognosis and higher mortality.

One of the advantages of our study is a homogeneous group of patients—only those diagnosed with HFrEF (LVEF < 40%) and treated for at least 3 months before the inclusion. Both the compensated and decompensated patients were enrolled, which allowed an observation that all the malnourished patients were hospitalized due to exacerbation of HF (100%). In the group at risk of malnutrition decompensation of HF occurred in 54%, and in patients with normal nutritional status only 32%, and these results were statistically significant (*p* = 0.002). Moreover, 74% of decompensated patients were malnourished or at risk of malnutrition. Treatment of CHF patients should combine conventional treatment with nutritional intervention to improve outcomes and reduce mortality [38].

The correlation between the nutritional status and LV function remains controversial, and the results vary in the available literature. There are studies using nutritional risk index (NRI) to evaluate nutritional status in HF patients with systolic ventricular dysfunction, which show that there is no correlation between nutritional status and LV function [39]. Similarly, in our study malnutrition was not correlated with echocardiographic parameters, including LVEF.

In our study, the majority of the malnourished patients demonstrated higher functional classes (NYHA III or IV), which confirms findings of other authors which show that malnourished patients have mostly NYHA III or IV [37]. There were no differences between the malnourished patients and those with normal nutritional status regarding hsCRP and natriuretic peptide concentrations, neither BNP nor NT-proBNP. However, a trend towards higher concentrations in the malnourished group was discernible for both BNP and NT-proBNP. Comparing the patients with normal nutritional status with the combined malnourished patients and those at risk of malnutrition showed significant differences as to NT-proBNP (*p* = 0.022). There are some studies showing higher CRP concentration in malnourished patients [37], which was not the case in our study. This may be caused by the homogeneity of our study population as opposed to the diverse patients enrolled in this other study [37].

Malnutrition may contribute to reduced exercise tolerance in patients with CHF. Our study showed that the malnourished patients demonstrated a significantly lower maximum oxygen consumption and higher VE/VCO_2_ slope as measured during cardiopulmonary exercise testing. Kazama et al. assessed nutritional status using another tool, GNRI (Geriatric Nutritional Risk Index) [40], but they similarly showed that peak VO_2_ was significantly lower in low GNRI group (15.8 ± 4.5 mL/kg/min vs. 18.3 ± 5.1 mL/kg/min, *p* < 0.001). Moreover, GNRI score was an independent determinant of low peakVO_2_ [40]. However, the investigated patients were somewhat older than those in our study (65.7 ± 12.8 years vs. 55 ± 11 years) [40]. The malnourished patients also demonstrated a shorter 6-min walking distance, which may indicate a risk of sarcopenia. The 6-minute-walking-test (6MWT) is one of the recommended tools for assessment of muscular strength in the European consensus statement on definition and diagnosis of sarcopenia [41]. In HF patients, sarcopenia may have a greater negative impact on tolerance of exertion and quality of life than cachexia [42]. Some authors state that HF patients tend to lose muscular tissue prior to fatty tissue, and that sarcopenia may contribute to development of or precede cachexia, which in turn is related to loss of muscular, fatty, and bone tissues [43].

Another angle was the research by Yasumura et al. The purpose of their study was to explore a simple prognostic indicator in patients with acute decompensated heart failure (ADHF) by including both nutritional status and physical capacity [44]. This study showed that malnutrition appeared in 49% of the analyzed population at admission to hospital, due to acute decompensated HF, and in 48% of these patients at discharge [44]. The authors reported that these patients had reduced physical capacity as measured in a simple test to determine if a patient could walk 200 m with a Borg scale score ≤ 13 and without critical changes in vital signs [44]. Malnutrition at discharge was more strongly related to mortality than that at admission by univariable analysis [44]. Neither malnutrition nor low physical capacity was related to heart failure rehospitalization by univariable analysis [44]. We reported that patients with malnutrition had worse activity level measured by 6 MWT, which is a better validated tool. The nutritional status did not influence survival during the follow-up, neither in singlevariate nor in multivariate analysis in our study. We have, nevertheless, used a different form to assess the nutritional status—MNA. Yasumura et al. used the geriatric GNRI. However, there are some differences between these two studies. In the Japanese study the population was much older than in ours (81 years vs. 58 years; respectively), and the LVEF was much higher (50% vs. 26.8%, respectively), which indicates great differences in the analyzed populations.

Serum albumin is a classical biomarker in HF related to nutritional status. The non-survivors showed lower albumin concentration (32.6 g/L vs. 47.4 g/L; *p* < 0.001). Hypoalbuminemia is common in patients with both stable and decompensated CHF and is independently associated with increased 1-year mortality in patients hospitalized due to HF decompensation [45]. However, we did not observe any correlations between albumin and protein concentrations and the nutritional status as measured by MNA. This suggests that albumin concentration cannot be used in diagnosis of malnutrition, though it remains a useful tool in assessment of prognosis in CHF patients.

There are some studies showing correlation between thyroid hormones and nutritional status. According to Asai et al., low fT3 level appears to be related to malnutrition and ageing in patients with acute HF, whilst in our study we did not observe such correlation, however we analyzed patients with CHF, so the analyzed group is different [46].

There are some biomarkers like serum cholinesterase, which has been used for the evaluation of nutritional status in daily practice [47] and its prognostic value was reported in patients with chronic HF (CHF) [48]. There are studies reporting the prognostic significance of serum cholinesterase level and superior predictive power of cholinesterase level to other objective nutritional indices, such as the controlling nutritional status score, prognostic nutritional index, and geriatric nutritional risk index in patients with acute decompensated HF [49]. Cholinesterase was a useful prognostic marker for prediction of adverse outcome in patients with HF with preserved ejection fraction/acute decompensated HF [49]. Despite being a useful biomarker, cholinesterase is not routinely measured in hospitalized HF patients.

This research shows several limitations. It is a single-center study, which limits the study sample. Hence its results should be interpreted with caution. We enrolled only HFrEF patients, but there were both stable and decompensated patients among them. This likely influenced the follow-up, and even though the nutritional status was adjusted for the severity of the disease, we cannot entirely rule out the sample heterogeneity bias. We used only one of the screening tools for the nutritional status evaluation, and did not compare it with the other tools. Yet it has to be emphasized that MNA seems to provide a reliable detection of malnutrition in patients with HF [50]. Additionally, the use of BMI can be controversial, as it does not reflect the body composition, especially considering the tendency of HF patients to accumulate body fluids (edema). Nevertheless, some researchers showed that BMI assessed in stable HF can be a useful indicator of mortality risk [51].

## 5. Conclusions

A considerable number of HFrEF patients show malnutrition or are at risk of malnutrition, which seems to be associated with worse outcomes and clinical status. The MNA score does not correlate with echocardiographic findings. Further research on a larger group of patients is needed to confirm the role of malnutrition in HF.

## Figures and Tables

**Figure 1 nutrients-12-02330-f001:**
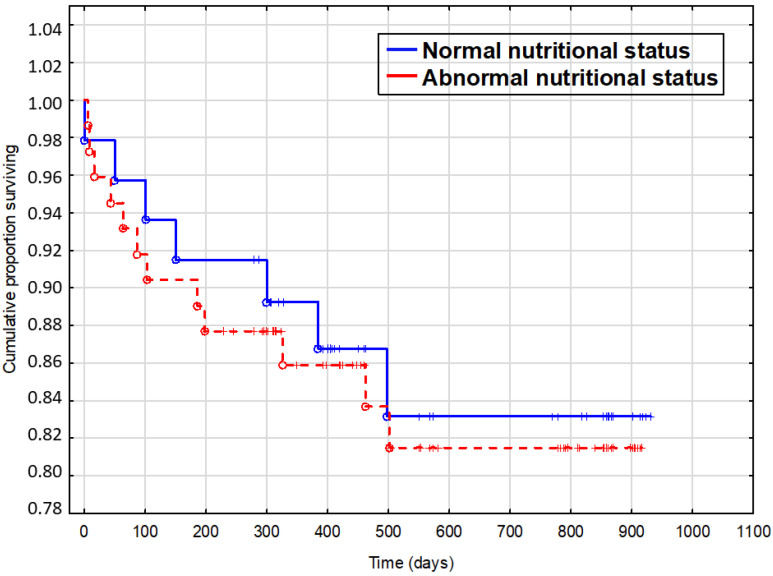
Influence of nutritional status on survival.

**Table 1 nutrients-12-02330-t001:** Clinical and laboratory characteristics of ‘normal nutritional status’, ‘at risk of malnutrition’, and ‘malnourishment’ subgroups.

	Normal Nutritional Status (*n* = 47)		At Risk of Malnutrition (*n* = 62)		Malnourishment (*n* = 11)		
	Mean	SD	Mean	SD	Mean	SD	*p*
Age [years]	54.02	11.92	56.26	10.64	50.55	13.71	0.51
Duration of hospitalization [days]	5.70	4.38	8.42	6.97	8.36	5.66	0.06
Rehospitalizations [number]	1.04	1.78	1.20	2.18	0.78	1.39	0.86
BMI [kg/m^2^]	28.66	4.86	28.79	5.81	22.18	6.64	<0.001
SBP * [mmHg]	110.87	17.84	108.15	16.14	96.00	8.43	0.018
HR ** [bpm]	71.12	15.64	75.98	15.77	83.20	13.21	0.009
NYHA	2.55	0.66	2.76	0.68	3.30	0.67	0.008
BNP [pg/mL]	649.73	781.30	740.00	739.04	1255.00	1029.46	0.18
NTproBNP [pg/mL]	2592.18	3700.82	3672.74	3365.66	4046.50	2383.70	0.06
TSH [uIU/mL]	2.92	2.69	2.97	4.83	2.80	3.39	0.51
fT3 [pg/mL]	3.21	0.63	3.14	0.79	2.91	0.44	0.59
fT4 [ng/dL]	1.41	0.27	1.48	0.30	1.60	0.49	0.61
CRP [mg/L]	7.62	9.75	9.05	10.53	13.50	8.54	0.05
ESR [mm/1 h]	14.74	11.44	12.29	11.08	11.33	6.25	0.74
Fe [umol/L]	14.12	5.70	14.39	6.37	11.67	5.69	0.76
Albumin [g/L]	44.64	13.86	45.19	15.73	46.30	19.67	0.92
TP [g/L]	69.19	12.95	70.03	12.97	56.26	22.40	0.09
Uric acid [umol/L]	426.73	158.28	453.38	156.88	467.60	127.02	0.36
Creatinine [umol/L]	111.01	45.85	112.23	34.41	95.35	28.11	0.42
eGFR [ml/min/1.73 m^2^]	72.45	25.67	68.10	23.12	73.55	21.17	0.58
Na+ [mmol/L]	138.43	3.31	137.67	3.97	137.27	3.82	0.57
K+ [mmol/L]	4.31	0.47	4.23	0.46	4.06	0.42	0.16
Fasting glucose [mmol/L]	6.01	1.05	5.98	1.67	6.02	1.97	0.47
HbA1c [%]	6.33	1.21	6.32	1.27	6.45	0.48	0.67
Bilirubin [umol/L]	18.92	12.68	19.33	15.15	40.00	40.64	0.55
AST [U/L]	41.09	54.54	49.59	62.87	87.18	161.00	0.18
ALT [U/L]	46.51	62.80	58.92	80.95	119.55	280.95	0.24
GGT [U/L]	94.71	71.52	118.73	117.01	174.43	107.87	0.24
Chol T [mmol/L]	4.67	1.61	3.96	1.23	3.61	0.90	0.021
LDL [mmol/L]	2.65	1.36	2.02	1.03	1.93	0.51	0.014
HDL [mmol/L]	1.39	0.51	1.32	0.47	1.18	0.40	0.53
TG [mmol/L]	1.38	0.84	1.35	0.72	1.29	0.53	0.99
RBC (×10e12)	4.52	0.51	4.82	0.70	4.54	0.66	0.05
HGB [mmol/L]	8.62	1.00	8.91	1.09	8.19	1.63	0.11
HCT [L/L]	0.41	0.04	0.43	0.05	0.40	0.07	0.14
WBC (×10e9/L)	7.49	1.79	8.31	2.26	7.13	1.34	0.08
PLT (×10e9/L)	193.23	67.49	197.77	61.28	243.45	110.26	0.32
LVEF [%]	26.80	9.95	25.31	9.89	23.41	16.67	0.16
LVEDD [mm]	70.43	11.62	70.38	11.30	64.91	13.47	0.59
LAD [mm]	52.41	9.36	54.35	10.29	47.09	14.31	0.17
RVD [mm]	36.93	6.80	36.92	6.45	36.73	4.36	0.99
PW [mm]	10.05	1.41	9.85	1.32	9.18	1.08	0.16
IVS [mm]	10.43	1.55	9.98	1.95	9.80	2.10	0.13
Ao [mm]	34.14	4.28	34.15	5.98	30.73	2.15	0.040
RVSP [mmHg]	38.71	13.58	40.50	11.78	39.67	3.51	0.47
TAPSE	15.94	4.89	13.95	4.26	17.00	1.41	0.17
6MWT [m]	408.17	88.10	340.76	127.35	208.33	98.02	0.031
pVO2 [ml/kg/min]	17.52	5.66	14.80	5.66	14.35	2.37	0.06
pVO2% [ml/kg/min]	55.76	16.27	49.50	15.31	50.25	5.44	0.35
pVO2 [L/min]	1.47	0.52	1.24	0.51	0.87	0.29	0.016
pVO2% [L/min]	53.14	16.95	45.64	13.86	44.25	4.92	0.20
VE/VCO2slope	31.99	8.92	37.03	8.14	39.13	6.68	<0.001
RER	1.05	0.09	1.03	0.09	0.93	0.12	0.17
Screening score	13.02	1.07	9.77	1.52	5.82	1.33	<0.001
Assessment	12.13	0.83	11.08	1.17	8.45	1.85	<0.001
Total assessment	25.15	0.89	20.85	1.77	14.27	2.64	<0.001

* on admission, ** at discharge, Abbreviations: BMI—body mass index, SBP—systolic blood pressure, HR—heart rate, NYHA—New York Heart Association, BNP—B-type natriuretic peptide, NT-proBNP—N-terminal pro B-type natriuretic peptide, TSH—thyroid-stimulating hormone, fT3—free triiodothyronine, fT4—free thyroxine, CRP—C-reactive protein, ESR—*erythrocyte sedimentation rate*, *Fe–iron*, TP—total protein, eGFR—estimated glomerular filtration rate, Na+-sodium, K+-potassium, HbA1c—hemoglobin A1C, AST—aspartate aminotransferase, ALT—alanine aminotransferase, GGT—gamma-glutamyltransferase, Chol T—total cholesterol level, LDL—low-density lipoprotein, HDL—high-density lipoprotein, TG—triglycerides, RBC—red blood cells, HGB—hemoglobin, HCT—hematocrit, WBC—white blood cells, PLT—blood platelets, LVEF—left ventricular ejection fraction, LVEDD—left ventricular end-diastolic diameter, LAD—left atrium diameter, RVD—right ventricular diameter, PW—posterior wall of left ventricle, IVS—interventricular septum thickness, Ao—aorta, RVSP—right ventricle systolic pressure, TAPSE—tricuspid annular plane systolic excursion, 6-MWT—6-min walking test, RER—respiratory exchange ratio.

**Table 2 nutrients-12-02330-t002:** Nonparametric characteristics of ‘normal nutritional status’, ‘at risk of malnutrition’, and ‘malnourishment’ subgroups.

	Normal Nutritional Status (*n* = 47)	At risk of Malnutrition (*n* = 62)	MALNOURISHMENT (*n* = 11)	*P* Value
Males	39 (83%)	53 (86%)	8 (73%)	0.57
Deaths	7 (15%)	9 (15%)	3 (27%)	0.55
HospitalizationHF	13 (28%)	11 (18%)	1 (9%)	0.27
NYHA 1-2	21 (48%)	20 (34%)	1 (10%)	0.06
NYHA 3-4	23 (52%)	39 (66%)	9 (90%)	
Non-ICM	30 (64%)	32 (52%)	7 (64%)	0.53
ICM	17 (36%)	30 (48%)	4 (36%)	
Exacerbation of HF	14 (30%)	30 (48%)	9 (82%)	0.004
Referred for HTX	8 (19%)	6 (10%)	3 (30%)	0.19
DM	7 (15%)	21 (34%)	4 (36%)	0.06
Insulin	3 (6%)	9 (15%)	1 (9%)	0.39
COPD	4 (9%)	7 (11%)	1 (9%)	0.88
CKD	8 (17%)	17 (27%)	1 (9%)	0.24
HA	18 (38%)	30 (48%)	4 (36%)	0.50
AF	23 (45%)	31 (50%)	2 (18%)	0.13
Thyroid disorders	11 (23%)	14 (23%)	3 (27%)	0.94
Stroke/TIA	2 (4%)	8 (13%)	0	0.16
Statin	20 (43%)	37 (61%)	4 (36%)	0.10
MRA	43 (91%)	57 (92%)	8 (73%)	0.13
Beta blockers	45 (96%)	61 (98%)	10 (91%)	0.34
ACEI/ARB/sacubitril +valsartan	45 (96%)	50 (81%)	8 (73%)	0.035

Abbreviations: HF—Heart Failure, NYHA—New York Heart Association, ICM—ischaemic cardiomyopathy, HTX—heart transplantation, DM—diabetes mellitus, COPD—chronic obstructive pulmonary disease, CKD—chronic kidney disease, HA—arterial hypertension, AF—atrial fibrillation, TIA—transient ischaemic attack, MRA—mineralocorticoid receptor antagonists, ACEI—angiotensin-converting enzyme inhibitors, ARB—angiotensin receptor blockers.

**Table 3 nutrients-12-02330-t003:** Differences between the patients with normal nutritional status and the combined groups of malnourished patients and those at risk of malnutrition.

	Normal Nutritional Status (*n* = 47)		At Risk of Malnutrition + Malnourishment (*n* = 73)		
	Mean	SD	Mean	SD	*p*
Age [years]	54.02	11.92	55.40	11.23	0.7675
Duration of hospitalization [days]	5.70	4.38	8.41	6.76	0.0192
Rehospitalizations [number]	1.04	1.78	1.14	2.09	0.7980
BMI [kg/m^2^]	28.66	4.86	27.80	6.36	0.4293
SBP * [mmHg]	110.87	17.84	106.29	15.79	0.2015
HR ** [bpm]	71.12	15.64	77.01	15.55	0.0124
NYHA	2.55	0.66	2.84	0.70	0.0422
BNP [pg/mL]	649.73	781.30	820.93	805.20	0.3517
NTproBNP [pg/mL]	2592.18	3700.82	3736.36	3200.38	0.0217
TSH [uIU/mL]	2.92	2.69	2.95	4.65	0.2638
fT3 [pg/mL]	3.21	0.63	3.11	0.75	0.4930
fT4 [ng/dL]	1.41	0.27	1.49	0.33	0.3740
CRP [mg/L]	7.62	9.75	9.72	10.33	0.1010
ESR [mm/1 h]	14.74	11.44	12.15	10.46	0.4456
Fe [umol/L]	14.12	5.70	14.13	6.27	0.8821
Albumin [g/L]	44.64	13.86	45.37	16.23	0.9852
TP [g/L]	69.19	12.95	67.66	15.68	0.7516
Uric acid [umol/L]	426.73	158.28	455.47	152.09	0.1919
Creatinine [umol/L]	111.01	45.85	109.68	33.91	0.4981
eGFR [ml/min/1.73 m^2^]	72.45	25.67	68.92	22.78	0.4614
Na+ [mmol/L]	138.43	3.31	137.61	3.92	0.3199
K+ [mmol/L]	4.31	0.47	4.20	0.46	0.1646
Fasting glucose [mmol/L]	6.01	1.05	5.98	1.70	0.2252
HbA1c [%]	6.33	1.21	6.34	1.18	0.8336
Bilirubin [umol/L]	18.92	12.68	22.11	21.05	0.7652
AST [U/L]	41.09	54.54	55.33	84.72	0.0810
ALT [U/L]	46.51	62.80	68.05	130.35	0.1140
GGT [U/L]	94.71	71.52	126.37	116.39	0.5471
Chol T [mmol/L]	4.67	1.61	3.91	1.19	0.0067
LDL [mmol/L]	2.65	1.36	2.00	0.97	0.0037
HDL [mmol/L]	1.39	0.51	1.30	0.46	0.4083
TG [mmol/L]	1.38	0.84	1.34	0.69	0.9370
RBC (×10e12)	4.52	0.51	4.78	0.70	0.0377
HGB [mmol/L]	8.62	1.00	8.80	1.20	0.4107
HCT [L/L]	0.41	0.04	0.42	0.05	0.2093
WBC (×10e9/L)	7.49	1.79	8.13	2.18	0.1329
PLT (×10e9/L)	193.23	67.49	204.66	71.70	0.3956
LVEF [%]	26.80	9.95	25.01	11.07	0.2147
LVEDD [mm]	70.43	11.62	69.54	11.73	0.8562
LAD [mm]	52.41	9.36	53.23	11.20	0.9564
RVD [mm]	36.93	6.80	36.89	6.15	0.9748
PW [mm]	10.05	1.41	9.74	1.30	0.1933
IVS [mm]	10.43	1.55	9.96	1.96	0.0628
Ao [mm]	34.14	4.28	33.62	5.69	0.5767
RVSP [mmHg]	38.71	13.58	40.42	11.26	0.2809
TAPSE	15.94	4.89	14.24	4.16	0.1291
6MWT [m]	408.17	88.10	320.90	130.47	0.0410
pVO2 [mL/kg/min]	17.52	5.66	14.75	5.41	0.0207
pVO2% [mL/kg/min]	55.76	16.27	49.58	14.59	0.1677
pVO2 [L/min]	1.47	0.52	1.20	0.50	0.0175
pVO2% [L/min]	53.14	16.95	45.50	13.20	0.0749
VE/VCO2slope	31.99	8.92	37.24	7.95	0.0028
RER	1.05	0.09	1.02	0.10	0.4960
Screening score	13.02	1.07	9.18	2.06	0.0000
Assessment	12.13	0.83	10.68	1.59	0.0000
Total assessment	25.15	0.89	19.86	3.04	0.0000

* on admission, ** at discharge, Abbreviations: BMI—body mass index, SBP—systolic blood pressure, HR—heart rate, NYHA—New York Heart Association, BNP—B-type natriuretic peptide, NT-proBNP—N-terminal pro B-type natriuretic peptide, TSH—thyroid-stimulating hormone, fT3—free triiodothyronine, fT4—free thyroxine, CRP—C-reactive protein, ESR—*erythrocyte sedimentation rate*, *Fe–iron*, TP—total protein, eGFR—estimated glomerular filtration rate, Na+-sodium, K+-potassium, HbA1c—hemoglobin A1C, AST—aspartate aminotransferase, ALT—alanine aminotransferase, GGT—gamma-glutamyltransferase, Chol T—total cholesterol level, LDL—low-density lipoprotein, HDL—high-density lipoprotein, TG—triglycerides, RBC—red blood cells, HGB—hemoglobin, HCT—hematocrit, WBC—white blood cells, PLT—blood platelets, LVEF—left ventricular ejection fraction, LVEDD—left ventricular end-diastolic diameter, LAD—left atrium diameter, RVD—right ventricular diameter, PW—posterior wall of left ventricle, IVS—interventricular septum thickness, Ao—aorta, RVSP—right ventricle systolic pressure, TAPSE—tricuspid annular plane systolic excursion, 6-MWT—6-min walking test, RER—respiratory exchange ratio.

**Table 4 nutrients-12-02330-t004:** Differences between the patients with normal nutritional status and the combined groups of malnourished patients and those at risk of malnutrition.

	Normal Nutritional Status (*n* = 47)	At risk of Malnutrition + Malnourishment (*n* = 73)	*P* Value
Males	39 (83%)	61 (84%)	0.93
Deaths	7 (15%)	12 (16%)	0.82
Hospitalization HF	13 (28%)	12 (16%)	0.14
NYHA 1-2	21 (47%)	21 (30%)	0.06
NYHA 3-4	23 (53%)	48 (70%)	
Non-ICM	32 (68%)	42 (58%)	0.50
ICM	15 (32%)	31 (42%)	
Exacerbation of HF	14 (30%)	39 (53%)	0.011
Referred for HTX	8 (19%)	9 (13%)	0.39
DM	7 (15%)	25 (34%)	0.019
Insulin	3 (6%)	10 (14%)	0.21
COPD	4 (9%)	8 (11%)	0.66
CKD	8 (17%)	18 (25%)	0.32
HA	18 (38%)	34 (47%)	0.37
AF	23 (49%)	33 (45%)	0.68
Thyroid disorders	11 (23%)	17 (23%)	0.97
Stroke	2 (4%)	8 (11%)	0.19
Statin	20 (43%)	41 (57%)	0.12
MRA	43 (91%)	65 (89%)	0.66
Beta blockers	45 (96%)	71 (97%)	0.85
ACEI/ARB/sacubitril + valsartan	45 (96%)	58 (79%)	0.012

Abbreviations: HF—Heart Failure, NYHA—New York Heart Association, ICM—ischaemic cardiomyopathy, HTX—heart transplantation, DM—diabetes mellitus, COPD—chronic obstructive pulmonary disease, CKD—chronic kidney disease, HA—arterial hypertension, AF—atrial fibrillation, TIA—transient ischaemic attack, MRA—mineralocorticoid receptor antagonists, ACEI—angiotensin-converting enzyme inhibitors, ARB—angiotensin receptor blockers.

**Table 5 nutrients-12-02330-t005:** Analysis of nutritional status.

*n* = 120	*p*-Value	Risk Ratio	Risk Ratio 95% Lower	Risk Ratio 95% Upper
Duration of hospitalization	0.228310	0.949402	0.872497	1.033085
SBP * [mmHg]	0.558559	0.986618	0.943074	1.032172
NYHA	0.128123	2.061004	0.811917	5.231740
NTproBNP [pg/mL]	0.449080	0.999938	0.999778	1.000098
TSH [uIU/mL]	0.453101	1.038639	0.940704	1.146771
CRP [mg/L]	0.224789	1.030790	0.981535	1.082516
Albumin [g/L]	0.007158	0.886242	0.811582	0.967770
eGFR [ml/min/1.73 m^2^]	0.014381	0.964413	0.936832	0.992805
HGB [mmol/L]	0.747388	0.928346	0.590499	1.459489
VE/VCO2 slope	0.102115	1.069292	0.986759	1.158728
Final score	0.812461	0.982084	0.845842	1.140272

* on admission, Abbreviations: SBP—systolic blood pressure, NYHA—New York Heart Association, NT-proBNP—N-terminal pro B-type natriureticpeptide, CRP—C-reactive protein, eGFR—estimated glomerular filtration rate, HGB—hemoglobin.

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
