# Peer review of "Relationship between Nutritional Status and Clinical and Biochemical Parameters in Hospitalized Patients with Heart Failure with Reduced Ejection Fraction, with 1-year Follow-Up"

_nutrients, 2020, doi:10.3390/nu12082330_

Round 1

Reviewer 1 Report

Overall this is a great manuscript assessing an important aspect in the management of heart failure. I have a few questions for the authors.

1) You had mentioned that none of the patients were on nutritional supplements? Can you please explain why despite being diagnosed with malnutrition in some of the patients, no steps were made to address the malnutrition. In the discussion part, the authors can give a brief explanation regarding what steps are taken to detect malnutrition at their institution and whether they have a nutritional team working along with the heart failure teams etc.

2) The authors have mentioned in the discussion that "in our study malnutrition was not correlated with echocardiographic parameters, including LVEF". Please specify if patients with malnutrition had increased incidence of other comorbidities like DM, CKD contributing to it.

 3) Page 12, line 264 and 265. This is an incomplete sentence " In that research 3 different malnutrition scores (PNI prognostic nutritional index, GRNI geriatric 264 nutritional risk index and CONUT controlling nutritional status).

4) Please check spacing in between words. In multiple places in the manuscript, 2 words are combined together without space in between.

Thank you.

Author Response

Dear Reviewer,

thank you for your valuable comments. We have made appropriate changes to the text. They were checked in change tracking mode. We added in the discussion suggested information.

The text has also been revised in terms of language.

Point 1: You had mentioned that none of the patients were on nutritional supplements? Can you please explain why despite being diagnosed with malnutrition in some of the patients, no steps were made to address the malnutrition. In the discussion part, the authors can give a brief explanation regarding what steps are taken to detect malnutrition at their institution and whether they have a nutritional team working along with the heart failure teams etc.

 Response 1: Malnutrition is an important and complicated problem in patients with HF, however it is difficult to diagnose inappropriate nutritional status and initiate effective treatment, particularly in HF patients. To diagnose malnutrition we use MNA form, which is detailed described in Methods, nonetheless we know it is not ideal method in this group of patients (e.g.: assessing BMI or calf or mid-arm circumferences in patients with oedema is contentious) – we are still developing methods of diagnosis and improving our knowledge and efficacy of treatment.   We cooperate with dietitian, who take care of our patients and help them understand their health/diet problem and try to solve it with the diet.

We have add it in the discussion part (line 318-325). 

Point 2: The authors have mentioned in the discussion that "in our study malnutrition was not correlated with echocardiographic parameters, including LVEF". Please specify if patients with malnutrition had increased incidence of other comorbidities like DM, CKD contributing to it.

Response 2: All differences between 3 subgroups (“normal nutritional status”, “at risk of malnutrition”, “malnourishment”) are presented in Table 1. (parametric characteristics) and Table. 2 (nonparametric characteristics). There were no correlation between malnutrition and incidence of any other comorbidities, such as e.g.: DM, COPD, CKD, HA, AF or cerebrovascular events (TIA/stroke) in the analyzed groups. However, we also divided patients into 2 subgroups: “normal nutritional status” and “inappropriate nutritional status”  (which was composed of malnourished patients and those at risk of malnutrition) and prepared statistical analysis presented in Table 3. (parametric characteristics) and Table 4. (nonparametric characteristics). This time we found association between inappropriate nutritional status and increased DM coexistence.

Our study only looked at HFrEF patients and in this population we found no association of LVEF with nutritional status.

 Point 3: Page 12, line 264 and 265. This is an incomplete sentence " In that research 3 different malnutrition scores (PNI prognostic nutritional index, GRNI geriatric 264 nutritional risk index and CONUT controlling nutritional status).

Response 3: This sentence should be completed: “In that research 3 different malnutrition scores (PNI prognostic nutritional index, GRNI geriatric nutritional risk index and CONUT controlling nutritional status) were used and none of them was MNA form.” It has already been changed in the manuscript body.

 Point 4: Please check spacing in between words. In multiple places in the manuscript, 2 words are combined together without space in between.

Response 4: It might be some technical problem, because before manuscript submission I had checked all spaces in between words. Nonetheless, it has been checked and changed again. Hopefully, there will be no any other typing mistakes.  

Reviewer 2 Report

Malnutrition or being at risk of malnutrition seems to be associated with both worse outcomes and clinical status in HFrEF patients.

I found this study interesting and well described.

I only suggest that the article is revised to avoid word fusion. In several passages, space between words is missing, specially when there is citations.

Author Response

Dear Reviewer,

thank you for your valuable comments. We have made appropriate changes to the text. They were checked in change tracking mode. We added in the discussion suggested information.

The text has also been revised in terms of language.

Point 1: Malnutrition or being at risk of malnutrition seems to be associated with both worse outcomes and clinical status in HFrEF patients.

I found this study interesting and well described.

Response 1: Thank you for your positive comments. We wanted to emphasize the problem of malnutrition in patients with HFrEF and show its clinical implications in this group of patients.

Point 2: I only suggest that the article is revised to avoid word fusion. In several passages, space between words is missing, specially when there is citations.

Response 2: It might be some technical problem, because before manuscript submission I had checked all spaces in between words. Nonetheless, it has been checked and changed again. Hopefully, there will be no any other typing mistakes.